# Activation of Persulfates Using Alkali-Modified Activated Coke to Promote Phenol Removal

**DOI:** 10.3390/nano15100744

**Published:** 2025-05-15

**Authors:** Yan Zhang, Shuang Shi, Jianxiong Wei, Qiang Ma, Xiaoxue Wang, Xingyu Zhang, Huarui Hao, Chen Yang

**Affiliations:** 1School of Chemistry and Chemical Engineering, Yulin University, No. 51 Chongwen Road, Yulin 719000, China; 15262719648@stu.yulinu.edu.cn (S.S.); wangxx@stu.yulinu.edu.cn (X.W.); hhr6886@sina.com (H.H.); yc000408@sina.com (C.Y.); 2Yulin Engineering Research Center of Coal Chemical Wastewater, Yulin University, No. 51 Chongwen Road, Yulin 719000, China; 3Yulin Innovation Institute of Clean Energy, Science and Technology Innovation 4th Road, Science and Technology Innovation New City, Yulin 719199, China; weijx@dnlyl.ac.cn; 4China Coal Shaanxi Energy & Chemical Group Co., Ltd., Yuheng Industrial Park, High-Tech Zone, Yulin 719099, China; 15353792879mq@sina.com; 5School of Materials Science and Engineering, Xi’an University of Technology, No. 5, Jinhua South Road, Xi’an 710048, China; zxy68168@sina.com

**Keywords:** alkali-modified activated coke, phenol, persulfate, removal, removal mechanism

## Abstract

Coke (AC) was modified and activated with sodium hydroxide (NaOH) and potassium hydroxide (KOH) to produce AC-Na and AC-K, respectively, and applied as a persulfate (PS) activator to promote phenol (Ph) removal in water. Under the given experimental conditions, compared to AC/PS (Ph removal effect was 77.09%), the Ph removal effects were 94.46% and 88.73% for AC-K/PS and AC-Na/PS, respectively. AC-K proved to be a more effective activator than AC-Na and was used for all the subsequent experiments. When PS/phenol molar ratio was 6.26:1:00, the initial system pH was 7 and the system temperature was 25 °C; the AC-K/PS system could effectively remove Ph (98.75%) from the simulated wastewater. After that, the stability of AC-K was verified. Electron paramagnetic resonance (EPR) and quenching analysis confirmed the hydroxyl free radical (•OH) to be predominant within this system. EPR combined with X-ray photoelectron spectroscopy (XPS), Fourier-transformed infrared (FTIR) spectroscopy, and Raman spectroscopy indicated that the sulfate radical (SO_4_^•−^) and •OH were generated due to the defects in AC-K, thereby enhancing the PS activation potency of AC-K. Additionally, the radical quenching experiments showed that the superoxide (O_2_^−^) radical is a key intermediate product promoting SO_4_^•−^ and •OH, which aided Ph removal. Both radical (SO_4_^•−^ and •OH) and non-radical (^1^O_2_) pathways were found to co-exist during the removal process. The Ph removal rate of the AC-K/PS system could still reach 29.50%, even after four repeated cycles. These results demonstrate that the unique AC-K/PS system has a potential removal effect on organic pollutants in water.

## 1. Introduction

Recent rapid industrial developments have increased to environmental pollution caused by organic waste [1]. Particularly, the organic matter produced during mining and smelting processes has severely impacted water bodies. Phenol (Ph; C_6_H_5_OH), a weak acidic aromatic compound consisting of a phenyl group (C_6_H_5_) and a hydroxyl group (OH) [2], has a sweet and distinguishable odor [3]. It is widely used in industries such as petrochemicals, pharmaceuticals, and coking [4,5,6]. Due to its high toxicity, phenol poses a risk to living organisms even at low concentrations (e.g., 9–25 mg/L for humans and animals [7]). The US Environmental Protection Agency (USEPA) classifies phenol as a priority pollutant and restricts its discharge into surface waters to concentrations below 1 ppb [8]. Effective methods for Ph removal from water, including adsorption, bioremoval, filtration, and chemical/electrochemical oxidation [9], are therefore essential.

Recently, chemical oxidation techniques have demonstrated significant potential for Ph removal [10]. For instance, persulfate (PS)-based advanced oxidation processes have been effectively removing organic contaminants from water for decades [11]. With a high redox potential (2.01 V), PS can be activated by ultraviolet (UV) light, heat, or transition metals to generate sulfate radicals (SO_4_^•−^, 2.5–3.1 V) for water treatment. Carbon materials such as activated carbon and biochar are economical, environmentally friendly, and non-toxic catalysts for activating PS to degrade organic matter [12,13]. However, their high costs remain a limitation.

The successful utilization of industrial and agricultural waste materials as pollutant scavengers in wastewater treatment has been reported. Activated coke (AC), with its porous structure, high specific surface area (SSA), large pore volume, abundant oxygen-containing functional groups, and low cost, has shown potential for advanced treatment of coking wastewater [14,15].

Surface modifications, including heat treatment [16], alkali modification [17], and nitration [18], have been applied to enhance the adsorption and oxidation capacities of carbon materials [19]. These techniques alter the surface functional groups and the pore size distribution of AC, thereby influencing its adsorption capacity and catalytic activity. Notably, AC modified by combined nitric acid (HNO_3_) and heat treatment exhibited the highest adsorption capacity and catalytic performance in sodium persulfate-mediated oxidation [20].

In this study, the effects of alkali modifications (via NaOH and KOH treatments) on the surface and chemical properties of activated coke (AC) in the AC/PS system were analyzed. The catalytic activities of alkali-modified AC for PS oxidation of phenol (as a model priority pollutant) were compared before and after surface modifications. Optimal experimental conditions and removal mechanisms were subsequently proposed.

## 2. Materials and Methods

### 2.1. Samples and Chemicals

We acquired Ph (purity 99.0%) from Tianjin Fuyu Fine Chemical Co., Ltd., Tianjin, China; coal-based AC from Shaanxi Sanlian Activated Coke Technology Co., Ltd., Yunlin, China; potassium persulfate (K_2_S_2_O_8_; ≥99.5%) from Tianjin Shengdian Chemical Reagent Co., Ltd., Tianjin, China; p-benzoquinone (BQ; ≥98.0%) from the China Pharmaceutical Reagent CP Hu Shi; and methanol (MeOH), tert-butanol (TBA), and other analytical reagents from Tianjin Kemiou Chemical Reagent Co., Ltd., Tianjin, China. All chemicals were of analytical grade and did not require further purification.

### 2.2. Alkali-Modified AC Preparation

The AC sample was initially crushed in a blender and sieved to a particle size of 200 mesh. The sieved samples were then dried in an oven, under closed conditions, at a temperature of 80 degrees centigrade, and stored in the dark. The dried AC was blended with KOH at a mass ratio of 1:2, and the mixture was heated in a nitrogen (N2) atmosphere, at a rate of 5 °C/min up to 750 °C, at which point it was maintained for 120 min before cooling back to room temperature. The resulting material was then acid-washed with hydrochloric acid (HCl), followed by rinsing with deionized (DI) water to neutralize it. The obtained dried potassium-modified AC sample (AC-K) was used to activate PS for Ph removal. Following the same procedure, we substituted KOH with NaOH to obtain sodium-modified AC samples (AC-Na). The method of AC activation via NAOH is consistent with the KOH method of activation, described above.

### 2.3. Experimental Process

#### 2.3.1. Batch Experiment

To investigate the Ph removal using modified AC (MAC)-activated PS, we conducted experiments comparing the MAC, PS, and MAC/PS systems (Ph concentrations = 30, 40, 50, and 60 mg/L; MAC dosage = 0.6 g/L; and PS dosage = 0.72 g/L) with pH adjusted at 25 °C. The Ph removal efficiency (R) by the AC- and MAC-activated PS can be calculated using Equation (1). All the experiments were performed in triplicate, and an average was presented.(1)$$R=\frac{C_{0}-C_{t}}{C_{0}}\times \;100\%,$$where C_0_ and C_t_ are the initial and t-time concentrations of Ph in the solution, respectively. Furthermore, the pseudo first-order kinetic equation used to determine the reaction rate constant (Kobs) for Ph removal by MAC-activated K_2_S_2_O_8_ under various conditions can be expressed as follows:(2)$$-\frac{dC}{dt}=K_{obs}C.$$
where Kobs is the reaction rate constant and C is the concentration of Ph.

Keeping all the other parameters constant and adjusting the considered parameter, the different influencing parameters on the removal efficiency of Ph were systematically investigated, by varying PS dosage (0.18–2.88 g/L), MAC dosage (0.2–1 g/L), initial pH (3, 5, 7, 9, 11), temperature (25 °C, 35 °C, 45 °C), and initial Ph concentration (30–60 mg/L).

For MAC recycling tests, the used MAC was washed alternatingly with deionized water and EtOH three times and then dried at 100 °C for 6 h. The sample was labeled as AC-K-De after the removal process.

#### 2.3.2. Radical Quenching Experiments

A total of 0.6 g of AC-K was added to 50 mL of the Ph solution (100 mg L^−1^). Subsequently, 50 mL of PS (1.44 g L^−1^) was introduced into the mixture. The reaction was conducted at 25 °C by adding the required MeOH, TBA, and BQ quenchers separately to the mixture. The pH of the solution was adjusted to 7 before placing it in a dual-function water bath thermostat shaker (SHZ-82, Xingtai Zhongde Machinery Manufacturing Co., Xingtai, China) set at 150 rpm. At reaction times of 0, 5, 10, 30, 60, 90, and 120 min, appropriate amounts of reaction solution were sampled, filtered using a suction filtration device (WIGGENS C300A, Eversheds Biotechnology Ltd., Shanghai, China), and transferred to a colorimetric tube. During the hydroxyl radical (•OH) and SO_4_^•−^ trapping experiment, a 1 mg/mL phenol solution was prepared. After ultrasonic dispersion (Ultrasonic POWERSONIC 520, Shanghai Zuo Le Instrument Co., Shanghai, China), 100 μL of this solution was first mixed with 200 μL of a 100 mM DMPO (5, 5-dimethyl-1-pyrroline-N-oxide) solution. Subsequently, 100 μL of a 10 mM PMS solution was added to the mixture and the timer was started. After 5 and 10 min, the samples were loaded into capillary tubes, sealed, placed into sample tubes, and tested for data collection in the machine (EPR A300-10/12, Bruker, Germany). A 1 mg/mL MeOH solution was then prepared via ultrasonication for the superoxide radical (•O_2_^−^) trapping experiment. We then mixed 100 μL of this solution with 200 μL of the DMPO solution, followed by 100 μL of the PMS solution, and started the timer. Tests were performed at 5 and 10 min, similarly to the previous case.

### 2.4. Characterizations

The specific surface, pore size distribution and pore volume of samples were carried out on a USA Micromeritics ASAP 2460 instrument using the nitrogen adsorption method. The X-ray diffraction (XRD; Advance D8, German Bruker) patterns of the samples were analyzed using a diffractometer with Cu KR radiation. The qualitative, quantitative, and chemical states of the elements before and after the reaction between the adsorbent and Ph solution were determined via X-ray photoelectron spectroscopy (XPS; Thermo Escalab 250 Xi; Micromeritics, Norcross, GA, USA). The aluminum (Al)-Kα X-ray source was operated at a voltage of 15 kV and a power of 12 mA. All binding energies were calibrated using carbon (C1S = 284.6 eV) as the reference. The morphologies of the samples were examined via scanning electron microscopy (SEM; Sigma 300 scanning electron microscope, Zeiss, Oberkochen, Germany) and energy dispersive X-ray spectrometer (EDS) analysis was conducted at the same time. The degree of graphitization or defect density of MACs were evaluated by Raman spectrum (Labram Hr Evolution; Horiba Scientific, Kyoto, Japan). Functional groups of the samples were investigated using Fourier transform infrared (FTIR) spectrometer (FTIR; Nicolet 6700, Germany Bruker, Wurzbach, Germany).

## 3. Result and Discussions

### 3.1. Characterizations Results

#### 3.1.1. BET Analysis Results

As can be seen from the N2 adsorption–desorption isotherms, the (BJH) and HK pore-size distribution curve (Figure 1), as well as the pore structure parameters of samples (Table 1), AC exhibited an SSA of 462.8 m^2^/g, with a small pore volume (Table 1). After KOH modification, the SSA of AC-K increased by up to 3.2 times (1468.8 m^2^/g), while the micropore area increased by 5.2 times of that of AC (1051.8 m^2^/g). Pore size analysis was conducted using the bjH method at 0–10 nm and the hk method at 0–2 nm for better observation of the pore size distribution of the materials in different ranges. Most AC-K pores were mesopores. Additionally, the total volume of AC-K (0.861 cm^3^/g) was 1.9 times that of AC, largely consisting of micropores, consistent with the N2 adsorption–desorption isotherms. The micropore volume of AC-K (0.527 cm^3^/g) reached 6.2 times that of AC.

For AC-Na, the SSA and total pore volume decreased slightly, from 462.8 to 426.3 m^2^/g and from 0.456 to 0.383 cm^3^/g, respectively, compared to AC. These results revealed a stronger effect of KOH on the AC pore structure optimization compared to NaOH; it provided more active surface sites and increased the activation of the PS system for better Ph removal effects.

These characteristics can be attributed to the following factors: potassium carbonate (K_2_CO_3_) was produced during the KOH modification of AC, which was thermally decomposed to potassium oxide (K_2_O), which further reacted with solid carbon to produce carbon monoxide (CO). This reaction played a critical role in pore formation. Moreover, the potassium (K) produced by the last reaction could also vaporize at high temperatures. The introduction of N_2_ made it easier for these vapors to enter the C layer for pore formation, increasing the SSA and micropore ratio of AC. The NaOH treatment of AC produced sodium carbonate (Na_2_CO_3_), which tends to be quite inert. These molecules accumulated in the pores, hindering AC pore access to N_2_ molecules. As a result, the SSA and pore volume of AC-Na were reduced.

#### 3.1.2. SEM

The sample surface images obtained via SEM technology revealed the AC surface to be relatively tight, with several pores (Figure 2). AC-K developed several irregular pores on the surface, indicating a corrosive effect of KOH modification, which increased the SSA and pore volume on the sample surface. The AC-Na surface showed minimal changes, confirming the conclusions of the pore structure distribution analysis. An elemental analysis of the samples revealed that compared to AC (O content = 10.63%; C content = 89.37%), the O content of (i) AC-Na decreased (5.13%) and that of (ii) AC-K increased (22.62%), while the C content of (i) AC-Na increased (94.80%) and that of (ii) AC-K decreased (76.37%).

#### 3.1.3. XRD

The XRD patterns revealed characteristic peaks of AC at 2θ = 22° and 43° corresponding to the (002) and (100) crystal planes of C, respectively (Figure 3) [21,22]. The appearance of diffraction peaks at 2θ = 22° indicated parallel stacking and inter-connection between portions of the graphite layer in AC, whereas the peak at 2θ = 43° indicated that sp2-hybridized C atoms in AC interactions form a hexagonal lattice structure. Such characteristic peaks could be found in the XRD patterns of AC-Na, but not AC-K. This indicated that the crystalline structure of AC might be damaged during the modified KOH process.

The diffraction peak intensities of the AC and the two AC samples modified by KOH and NaOH varied to different extents (Figure 4). The two diffraction peaks of AC-Na [Figure 4c,d] were sharper and stronger than those of AC [Figure 4a,b] and AC-K [Figure 4e,f]. The latter experienced a clear diffraction peak intensity reduction, possibly due to KOH dismantling the graphite crystal structure of AC. We obtained fits of the (002) and (001) crystal planes for all samples with the peakfit software(version number 4.12), corresponding to diffraction peaks at 2θ = 26.6° and 43.4°, respectively.

The (002) plane represented the microcrystalline structure formed by the condensation of the aromatic nucleus. According to the fitting results, we computed different parameters of the three carbon crystalline structures (Table 2), including d002, Lα, Lc, and N, expressed as follows:(3)$$d_{002}=\frac{\lambda }{2sin\;\theta_{002}},$$(4)$$L_{\alpha }=\frac{1.84\lambda }{B_{100}cos\;\theta_{100}},$$(5)$$L_{c}=\frac{0.89\lambda }{B_{002}cos\;\theta_{002}},$$(6)$$N=\frac{L_{c}}{d_{002}},$$
where λ is the wavelength of the incident X-ray (1.54 Å) and B002 and B100 are (002) on the XRD pattern, respectively. The half-maximum width corresponding to the peak and (100) peak, rad; and θ002 and θ100 correspond to the (002) and (100) peaks in the XRD patterns, respectively. d002: interlayer spacing; Lα: average size; Lc: stacking height; N: tier number.

Compared to AC, d002 increased and Lα, Lc, and N decreased for both AC-K and AC-Na. These effects could occur due to the alkali modifiers dismantling the carbon microcrystalline structure of AC, increasing the aromatic layer spacing and reducing its size and degree of crystallization order. Moreover, the substantial modification effects of KOH over NaOH indicated two-dimensional damage of KOH to aromatic crystallines. The rise in d002 increased the SSA and absorption capacity [23,24], which was consistent with the BET analysis results.

#### 3.1.4. FTIR

The FTIR spectra of all three materials showed a broad and intense absorption band at 3800–3200 cm^−1^, attributed to the hydroxyl group (O-H) stretching vibrations (Figure 5). Aliphatic C-H stretching vibration peaks were observed at 3000–2700 cm^−1^. Moreover, absorption bands of graphite (C-C) and O-containing functional groups (C=O and C-O) were observed at 1800–800 cm^−1^ [25,26].

The absorption bands of AC-K were similar to those of AC-Na. However, compared to AC, the strength of the above characteristic peaks reduced significantly post modifications. Weakening of the peak at 4000–3200 cm^−1^ indicated hydrolysis of the OH group. The antisymmetric stretching vibration of the C=O bond at 1600 cm^−1^ and the C-O stretching vibration of the unsaturated ether bond at 1200 cm^−1^ also weakened due to the pyrolysis process, indicating that the C and O content in AC decreased with the increasing temperature and activation medium. Additionally, a new peak emerged at 1058 cm^−1^, which was attributed to C-O stretching vibrations, indicating the presence of tertiary OH, which is beneficial for PS activation and enhanced Ph removal. The aforementioned results indicated that after modifications, the proportion of H and O in AC-K and AC-Na decreased, while the organic components carbonized, which was in line with the EDS analysis results.

#### 3.1.5. Raman Spectroscopy

For all samples, two characteristic peaks centered at 1350 and 1600 cm^−1^ were observed, corresponding to the D and G bands, respectively, (Figure 6). Typically, the D band represents defects and disorders in the material, while the G band reflects the E2g vibration of sp2-hybridized graphitic C atoms, confirming graphitization. The defects in the structure can effectively activate PS, whereas the presence of graphite aids electron transfer and accelerates the catalytic ability. The structural characteristics of the material can be seen from the strength ratio of D bonds and G bonds [27].

The AC-K sample exhibited a higher ID/IG (1.030) compared to AC (0.859) and AC-Na (0.938). Furthermore, the AC-K-De results revealed a reduction in graphite content after Ph removal. The graphitic defects and edge defects of O-containing groups in AC-K can serve as effective catalysts to promote advanced PS oxidation, thereby accelerating Ph removal.

#### 3.1.6. XPS

XPS was employed to further explore the state of AC, AC-K, and AC-Na in detail. C and O were detected on the surface of AC (Figure 7). As expected, the spectra of AC-K and AC-Na additionally detected the presence of K and Na, respectively. K is known to mainly exist in the form of O2/KC8 and KC8, generated by the reaction of KOH with AC [28].

As shown in Table 3, compared to AC, the relative content of element C for AC-K decreased from 91.91% to 51.34%, while the relative content of O increased from 8.09% to 47.49%. These results indicated that KOH underwent a redox reaction with C in AC, thereby reducing (increasing) the surface C (O) content [29,30]. However, the relative O and C content in AC-Na remained the same as in AC.

As seen in Figure 8 and Table 4, the high-resolution O1S spectra of AC could be split into four peaks of 531.03, 532.21, 533.30, and 534.10, corresponding to C=O (carbonyl group), C=O (ester, amides), C-OH, and -COOH bands, respectively. The C=O and C-HO bands decreased from 29.03% and 41.00% in AC to (i) 8.30% and 14.70% in AC-K, respectively and (ii) 3.00% and 15.8%, respectively, which was consistent with the FTIR results.

After the reaction of the AC-K/PS degraded Ph, C–O–C bonds decreased from 61.5% to 25.3%, due to the decomposition of Ph molecules through nucleophilic attack or electron transfer [31]. Additionally, O–C=O and C=O bonds increased from 15.5% and 8.3%, respectively, to 20.7% and 23.6%, respectively. These changes indicated that during Ph removal, the redox reaction reduced the C–O–C content and increased the carbonyl and carboxyl functional groups, with significant transformations to the surface structure (Table 5).

The high-resolution C1S spectra of AC could be split into three peaks of 284.7, 286.1 and 287.9 eV, corresponding to primary *sp*^2^-C/C=C, C–OH, and C=O bonds, respectively. In AC-K, C=O and primary *sp*^2^-C/C=C bonds increased from 13.1% and 78.5% to 19.2% and 65.0%, respectively. the C=O bonds increased to 28.1% and the primary *sp*^2^-C/C=C bonds decreased to 46.6% after the reaction. The results were in line with the analysis of FTIR and Raman spectra (Figure 9 and Table 6).

### 3.2. Determination of Alkali Modifier

The removal of Ph by AC/PS, AC-K/PS, and AC-Na/PS systems was investigated to select the best modifier. Under the given experimental conditions, the Ph removal of AC-K and AC-Na was improved by 123% and 106%, respectively, compared with that of AC (Figure 10). The best results of AC-K were attributed to its much larger specific surface area and excellent pore structure, which provided more surface active sites. In addition, the number of oxygen-containing functional groups in AC-K that act as electron donors for activating the PS to generate SO_4_^•−^ was also increased. Therefore, in the subsequent experiments, we chose KOH as the modifier because of its superior performance, compared to NaOH.

### 3.3. Factors Influencing Ph Removal of MAC/PS Systems

#### 3.3.1. PS Dosage

Experimental data on Ph removal efficiency indicated that in the absence of PS, AC-K alone showed no appreciable absorption for Ph (Figure 11). AC-K with a graphitized structure could generate π–π electron donor-acceptor interactions with Ph. As the Ps dosage increased from 0.18 to 0.72 g/L, the Ph removal efficiency increased from 81.6% to 88.5% within 120 min. This enhancement could be attributed to higher PS dosages, which could facilitate the generation of reactive oxidative species (ROS).

However, when the PS dosage furtherly increased to 2.88 g/L, the Ph removal efficiency decreased. This result was attributed to a competitive adsorption between S_2_O_8_^2−^ and phenol on the AC-K surface as the amount of PS is overdoes [32]. Furthermore, S_2_O_8_^−^ were generated through the reaction of the excess S_2_O_8_^2−^ with SO_4_^•−^ [33,34], leading to a reduction in the phenol removal rate. Thus, 0.72 g/L was selected as the optimal PS dosage for the subsequent experiments.

#### 3.3.2. AC-K Dosage

In the absence of AC-K, PS alone exhibited a Ph removal efficiency of 8.76%. Increasing the AC-K concentration from 0.2 to 0.6 g/L in the AC-K/PS system raised the efficiency from 58.48% to 83.74%. A further increase in AC-K dosage resulted in a negligible enhancement, suggesting 0.6 g/L to be the appropriate AC-K concentration for providing the maximum number of active sites for efficient Ph removal (Figure 12).

#### 3.3.3. Effect of pH

The effect of the initial pH value on the Ph removal efficiency was investigated over the range of 3–11 (Figure 13). The highest removal efficiency was observed at Ph 7.0. As the initial pH increased from 7 to 11, the Ph removal efficiency decreased from 89.35% to 66.65%. These results indicated a strong pH dependence of the removal reaction in the AC-K/PS system due to the various forms of the pollutants at different Ph values and the surface charge properties of the activator [35]. Thus, 7 was selected as the optimal pH value for the subsequent experiments. The experiments showed that the removal efficiency of the system exhibited a significant pH dependence: it reached its peak at pH = 7 (89.35%), and then gradually decreased to 66.65% with the increase in pH to 11. The phenomenon originated from a multifactorial synergistic mechanism: (1) alteration of pollutant ionization equilibrium: the target pollutants were less protonated under alkaline conditions, leading to a decrease in their aromatic ring electron cloud density, which weakened the π-π electron transfer efficiency with the active site of the catalysts; (2) charge inversion on the surface of the carriers: the charge characteristics near the isoelectric point of the activated carbon surface (pI ≈ 6.8) dominated the adsorption behaviors, and the negative surface charge density increased and inhibited the adsorption behavior when the pH was higher than 7. When pH > 7, the surface negative charge density increased, inhibiting the adsorption of negatively charged pollutants.

#### 3.3.4. Effect of Temperature

The Ph removal efficiency of the AC-K system after 120 min at 25 °C, 35 °C, and 45 °C was 79.80%, 78.88%, and 88.50%, respectively, (Figure 14). Therefore, increasing the temperature improved the Ph removal efficiency. This effect was due to thermal activation accelerating the decomposition of PS to produce more ROS and higher temperatures promoting collisions between AC-K and ROS [36,37].

However, as the Ph removal efficiency enhancement caused by higher temperatures was relatively insignificant, we conducted further experiments at 25 °C, with minimal system temperature adjustments, in order to conserve energy.

#### 3.3.5. Effect of Initial Concentration of Ph

The effect of the initial concentration of Ph on the removal efficiency was illustrated in Figure 15. The Ph solution was completely degraded in 30 min and the 30.00 mg Ph solution was completely degraded in 90 min. When the initial concentration of Ph was increased to 50, and the removal rate was 94.488%. within 120 min. The initial concentration of ph was further increased to 60 mg/L, and its removal rate was 96.875% within 120 min. For the given AC-K/PS system, the amount of reactive free radicals generated in the solution was considered to be constant, and was completely consumed. In addition, high concentration of Ph would deplete the AC-K which made the removal efficiency decrease. Thus, the further experiments were performed at an initial concentration of 40 mg/L Ph.

### 3.4. Reactivity and Reusability of MAC

The activity of recycled AC-K for Ph removal was investigated under the aforementioned optimal experimental conditions, with an initial Ph concentration of 40 mg/L (Figure 16). The AC-K was washed six times, alternating between DI water and EtOH, and calcinated at 110 °C for 12 h after each use. Although a small amount of AC-K was lost during the cleaning process, the Ph removal rate for the fourth cycle was maintained at 29.50%, indicating common chemical stability and reusability in a practical application.

### 3.5. Identification of Primary ROS

#### 3.5.1. Radical Scavenging

The radical scavenging experiments used MeOH and TBA as efficient radical quenching agents. MeOH can scavenge both SO_4_^•−^ (1.1 × 10^7^ M^−1^s^−1^) and •OH (9.7 × 10^8^ M^−1^s^−1^), while TBA is often employed as the dominant scavenger of •OH (6 × 10^8^ M^−1^s^−1^) [38].

As 100 mM MeOH or TBA was added to the system, the Ph removal efficiency of AC-K decreased from 94.96% to 87.76% and 88.15%, respectively (Figure 17). The inhibition effect of MeOH and TBA showed that •OH and SO_4_^•−^ coexisted in the MAC/PS system, with the former expected to be the dominant free radical species [39]. Nevertheless, Ph removal was not completely inhibited, indicating the participation of other reactive species in the oxidation process.

#### 3.5.2. EPR Analysis

Carbon activation of PS has been reported to effectively produce singlet oxygen (O_2_^•−^) [40]. The EPR analysis with DMPO and TEMP as the radical spin-trapping agents were performed to further analyze the O_2_^•−^ generated in the AC/PS system. As in Figure 17c,d, we detected a clear DMPO–OH (αH = αN = 14.8 G hyperfine splitting constant, 1:2:2:1 quartet) and weak DMPO–SO_4_ (αN= 13.84 G, αH =10.1 G, αH = 1.46 G, and αH = 0.80 G) signals when using DMPO as the spin trapping agent [41], consistent with the radical quenching experiments. Moreover, the intense triplet signals with the equal intensity ratio of TEMP–O_2_ [Figure 17d], indicated the presence of O_2_^•−^. The above results verified the presence of •OH, SO_4_^•−^ and O_2_^•−^, where the •OH (radical pathway) and ^1^O_2_ (non-radical pathway) predominated the removal process.

### 3.6. Reaction Mechanism

Due to the particularly dominant role of •OH in the AC-K/PS system, we speculated that it originates from H2O or OHˉ. However, PS cannot directly convert H_2_O to •OH [42], and PS itself cannot effectively remove Ph. In their research, Ahmad et al. [43] show that persulfate (PS) is doubly activated under highly alkaline conditions (pH > 10.5) via a synergistic mechanism of alkaline hydrolysis (generating HO_2_^−^) and phenolate single-electron reduction (generating SO_4_^−^), and that its superimposed effect significantly enhances the efficiency of free radical generation, which provides a key mechanistic support for the optimization of persulfate activation strategies in environmental remediation. However, as our experimental conditions were controlled at pH = 7, we ruled out the possibility of alkali and phenolic compounds being activated. Therefore, it is reasonable to believe that the activation of PS originates from the inherent characteristics of AC-K.

According to the results and analysis, the intrinsic catalytic mechanism of AC-activated PS for Ph removal was established (Figure 18). Briefly, the ketone groups attached to the AC are known to be crucial functional groups generating ^1^O_2_ [44]. As removal progressed, the ID/IG ratio increased, indicating that the graphitized carbon on the AC materials could serve as the active site. The graphitized structure implied the presence of ketone groups in AC-K, which could activate PS to produce reactive ^1^O_2_ during the Ph removal process. Furthermore, the self-decomposition of PS and defects in AC-K might also be responsible for the non-radical pathway in the AC/PS system [45,46].

Furthermore, PS molecules could be initially anchored onto the AC-K surface, after which its sp2-hybridized structure activated PS, rendering it metastable by changing its electron configuration through interaction with the O–O bonds of PS. With the reaction beginning, Ph is an electron-rich pollutant, while AC-K can be treated as a mediator to accelerate the electron transfer between Ph molecules and metastable PS.

According the previous studies, the surface functional groups of carbon materials were considered to have a critical role in their catalytic performance [47]. XPS spectra of the AC-K surface revealed the presence of O-containing functional groups (e.g., C–OH and C=O), as the electron donors activating PS to generate SO_4_^•−^ and oxidize C=C groups to produce more C=O groups. Additionally, dangling bonds can be generated by the edge defects of AC-K, making their elections out of those confined by edge carbons, thereby allowing AC-K to provide electrons to PS to form •OH and SO_4_^•−^ (Equations (7)–(10)). In addition, SO_4_^•−^ can react with H_2_O to rapidly yield •OH. Overall, there are two pathways, the radical and the non-radical, in the AC-K/PS system (Equation (11)).

The high phenol removal efficiency of AC-K/PS may be attributed to its large specific surface, which promotes the physical adsorption of phenol and S_2_O_8_^2−^. Consequently, the simultaneous generation of radical and non-radical species, i.e., •OH, SO_4_^•–^, and ^1^O_2_, may oxidize the phenol absorbed around the AC-K to inorganic substances such as CO_2_ and water (Equation (12)) [47].(7)$${SC}_{surface}-C=O+S_{2}O_{8}^{2-}\to {SC}_{suface}-{CO}^{•}+{{SO}_{4}}^{•-}+{SO}_{4}^{2-}$$(8)$${SC}_{surface}-OH+S_{2}O_{8}^{2-}\to {SC}_{suface}-O^{•}+{{SO}_{4}}^{•-}+{HSO}_{4}^{-}$$(9)$${SC}_{defect}+S_{2}O_{8}^{2-}\to {SC}_{defect}^{+}+{{SO}_{4}}^{•-}+{SO}_{4}^{2-}$$(10)$${SO_{4}}^{•-}+{OH}^{-}+{O_{2}}^{•-}⟶{SO_{4}}^{2-}+•OH$$(11)$${SO_{4}}^{•-}+H_{2}O⟶{SO_{4}}^{2-}+•OH+H^{+}$$(12)$$HO•+{SO_{4}}^{•-}+C_{6}H_{5}OH⟶Severalsteps⟶CO_{2}+H_{2}O+{SO_{4}}^{2-}$$

## 4. Conclusions

In this study, PS was successfully activated by KOH-modified AC (i.e., AC-K) to achieve efficient removal of Ph. Simultaneously, KOH modification disrupted the graphite crystal structure of AC and generated more structural defects, providing sufficient sites for PS activation.

Under the ideal conditions for the AC-K/PS system (i.e., PS/Ph molar ratio = 6.26; initial system pH = 7; system temperature = 25 °C; and initial Ph concentration = 40 mg/L), the Ph removal rate reaches 97.31% in 120 min. AC-K demonstrates excellent initial performance with a Ph removal efficiency of 98.75% when first used, and has a significant advantage in recyclability: it retains a removal capacity of 29.50% through four cycles, effectively reducing the cost of use and the environmental burden, and achieving highly efficient pollutant removal, especially in short-term application scenarios. However, the material has obvious deficiencies in cycling stability, with a cumulative efficiency loss of 70.13% (69.25 percentage points in absolute terms) after four cycles, and an average efficiency loss of 17.31% in a single cycle, suggesting a significant decrease in performance after long-term repeated use. Quenching experiments and EPR tests implied that the simultaneous generation of radical and non-radical species, i.e., •OH, SO_4_^•−^, and ^1^O_2_, validated them as dominant reactive oxygen species (ROS) in charge of removing phenol in AC-K/PS system. ROS, mainly derived from the AC-K structures, contains defect edges, graphized carbon, and oxygen-containing functional groups. In brief, the application of alkali-modified AC can be utilized as an effective PS activator for environmental remediation.

## Figures and Tables

**Figure 1 nanomaterials-15-00744-f001:**
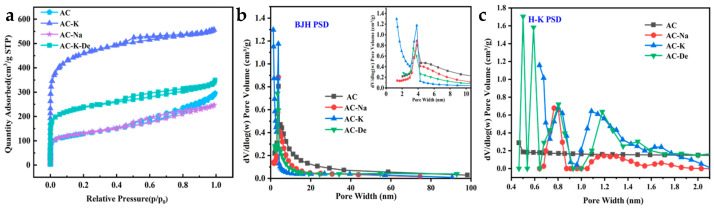
The N_2_ adsorption–desorption isotherms of activated coke raw powder, NaOH-activated AC-Na, KOH-activated AC-K, and AC-k after the phenol removal were measured at a liquid nitrogen temperature of 77 K. (**a**) The adsorption–desorption isotherm; (**b**) Barrett–Joyner–Halenda (BJH) pore size distribution; and (**c**) Horvath–Kawazoe (H-K) pore size distribution.

**Figure 2 nanomaterials-15-00744-f002:**
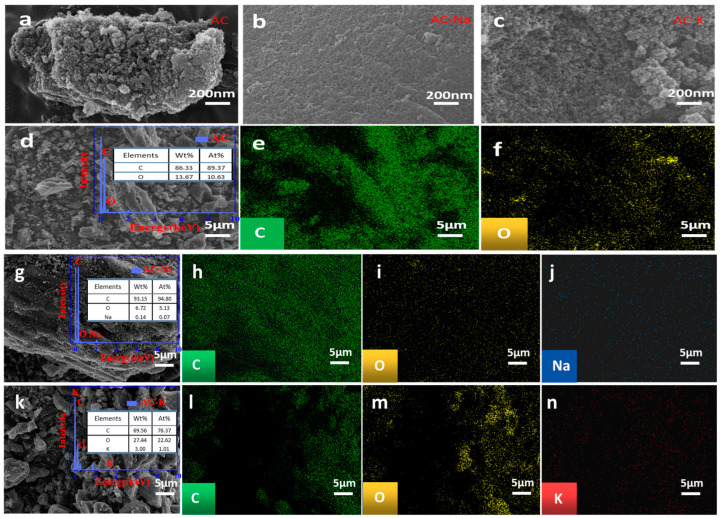
(**a**–**c**) SEM images of AC, AC-Na, and AC-K, respectively; (**d**) energy dispersive X-ray spectroscopy (EDS), element weight percentage (wt.%), and atomic percentage (at.%); (**e**,**f**) C and O distribution on AC surface; (**g**) AC-Na element wt.% and at.%; surface scanning of (**h**) C, (**i**) O, and (**j**) sodium (Na) content in AC-Na; and (**k**) AC-K element wt.% and at.%; and surface scanning of (**l**) C, (**m**) O, and (**n**) K in AC-K.

**Figure 3 nanomaterials-15-00744-f003:**
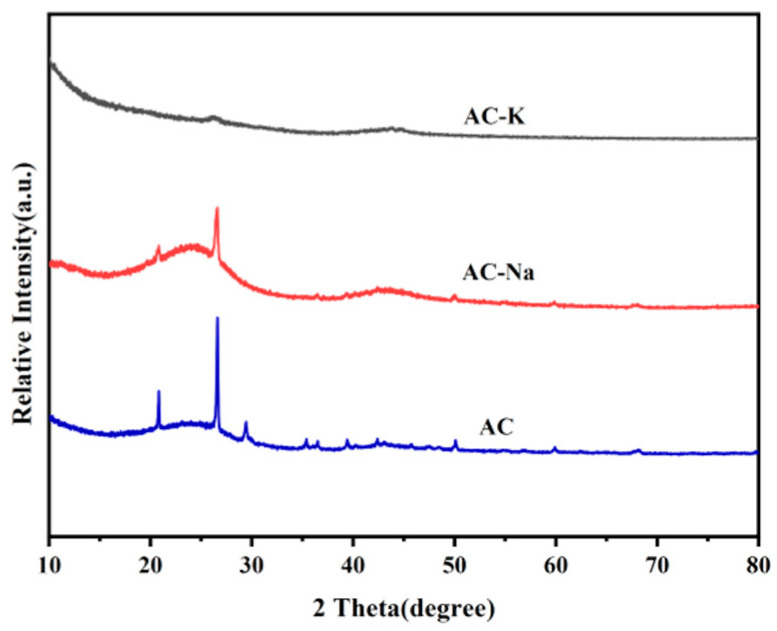
XRD patterns of AC, AC-Na, and AC-K.

**Figure 4 nanomaterials-15-00744-f004:**
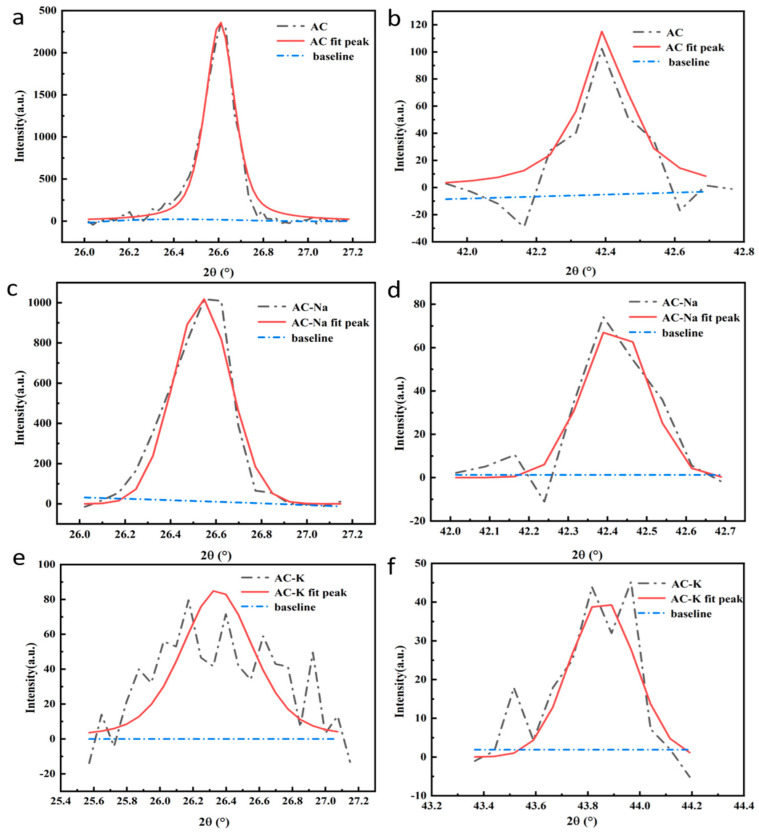
XRD peak intensity fitting of (**a**,**b**) AC, (**c**,**d**) AC-Na, and (**e**,**f**) AC-K at (**a**,**c**,**e**) 002 (2θ~26.6°) and (**b**,**d**,**f**) 100 (2θ~43.4°) crystal planes.

**Figure 5 nanomaterials-15-00744-f005:**
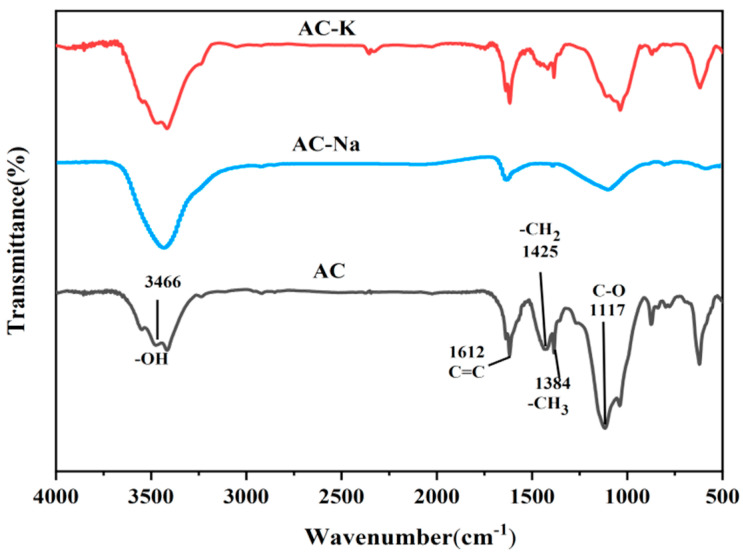
FTIR spectra of AC, AC-K, and AC-Na.

**Figure 6 nanomaterials-15-00744-f006:**
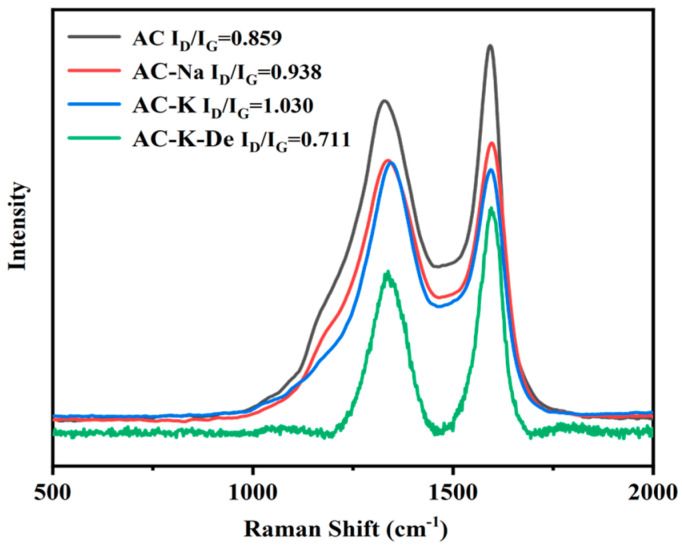
Raman spectra of AC, AC-K, AC-Na, and AC-K-De.

**Figure 7 nanomaterials-15-00744-f007:**
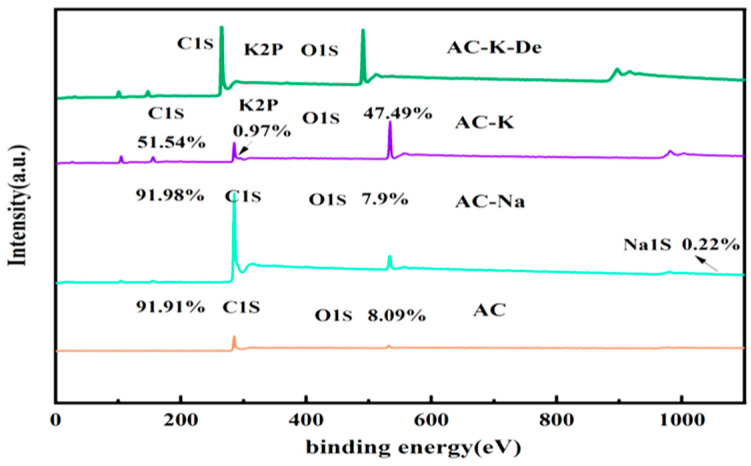
XPS spectra of AC, AC-K, AC-Na, and AC-K-De.

**Figure 8 nanomaterials-15-00744-f008:**
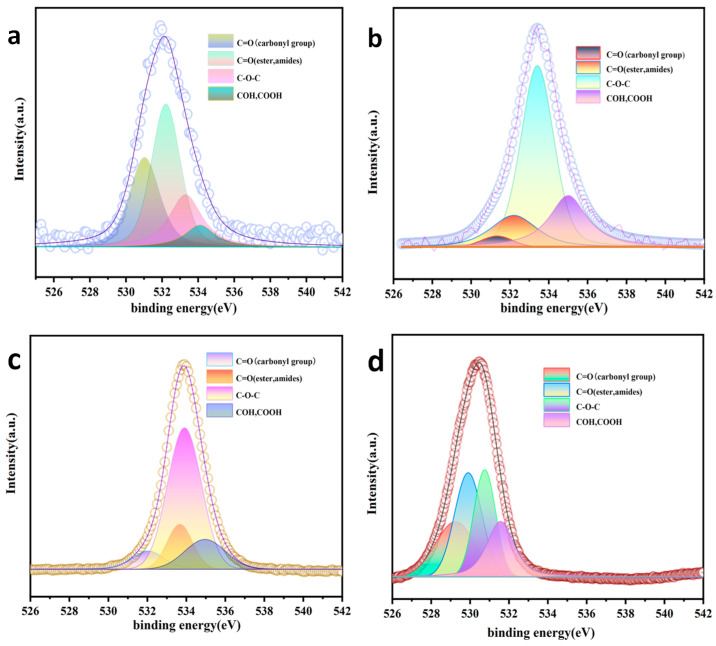
XPS spectra of O1S groups in (**a**) AC, (**b**) AC-K, (**c**) AC-Na, and (**d**) AC-K-De.

**Figure 9 nanomaterials-15-00744-f009:**
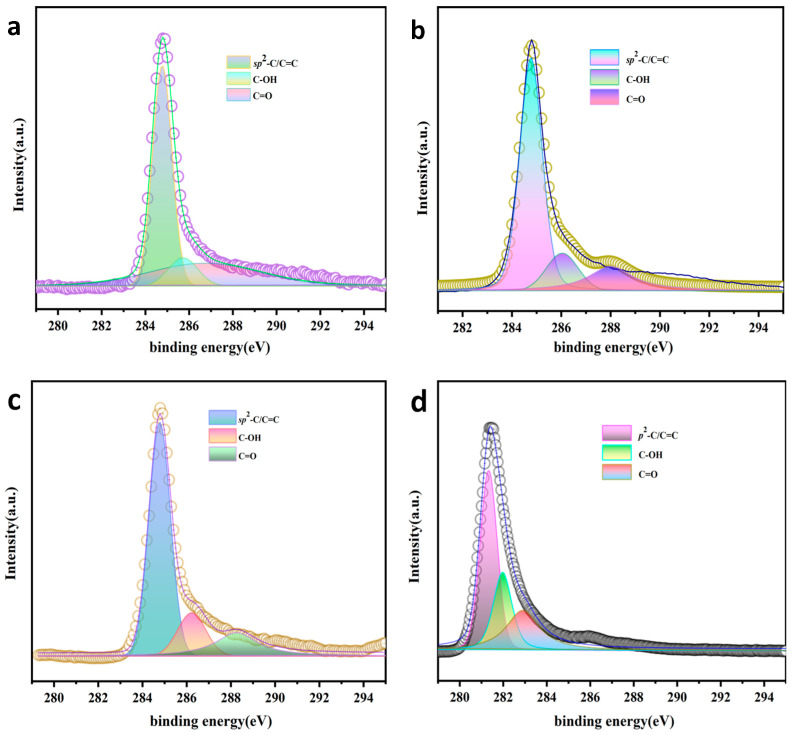
XPS spectra of C1S groups in (**a**) AC, (**b**) AC-K, (**c**) AC-Na, and (**d**) AC-K-De.

**Figure 10 nanomaterials-15-00744-f010:**
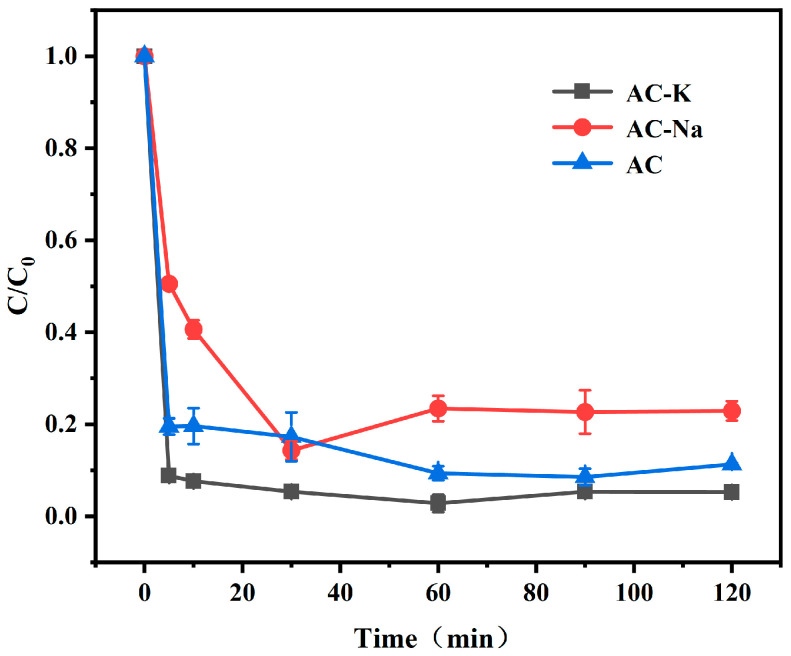
Removal of Ph from aqueous solutions by AC-K, AC-Na, and AC (reaction condition: MAC = 0.6 g/L, Ph = 40 mg/L, PS = 1.44 g/L, pH = 7, T = 25 °C).

**Figure 11 nanomaterials-15-00744-f011:**
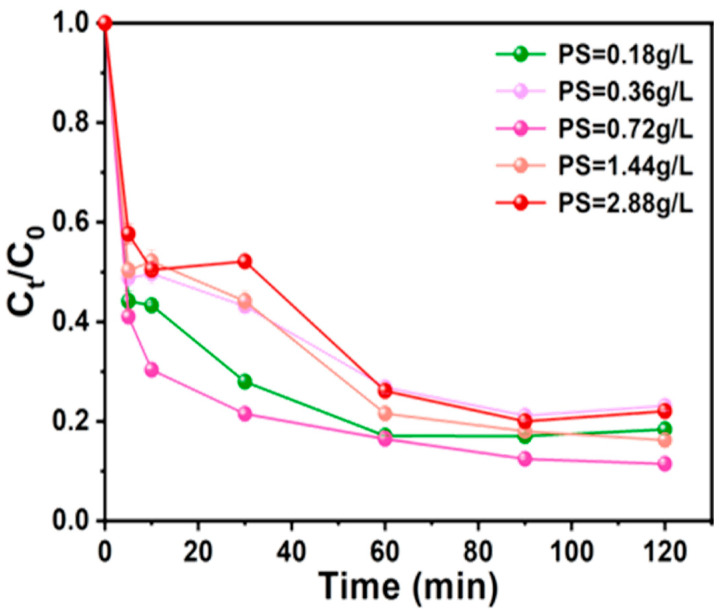
Effect of PS dosage on Ph removal (Ph = 50 mg/L, MAC = 0.6 g/L, T = 25 °C, pH = 7).

**Figure 12 nanomaterials-15-00744-f012:**
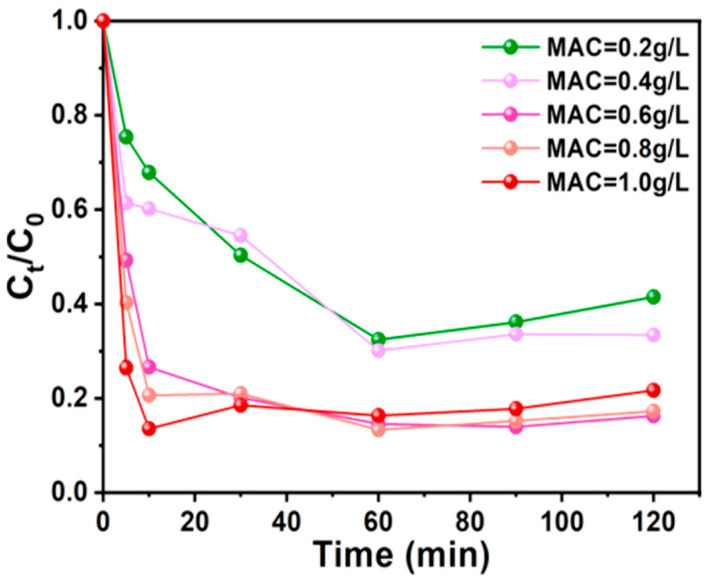
Effect of AC-K dosage on Ph removal (Ph = 50 mg/L, MAC = 0.6 g/L, PS = 0.72 g/L, T = 25 °C).

**Figure 13 nanomaterials-15-00744-f013:**
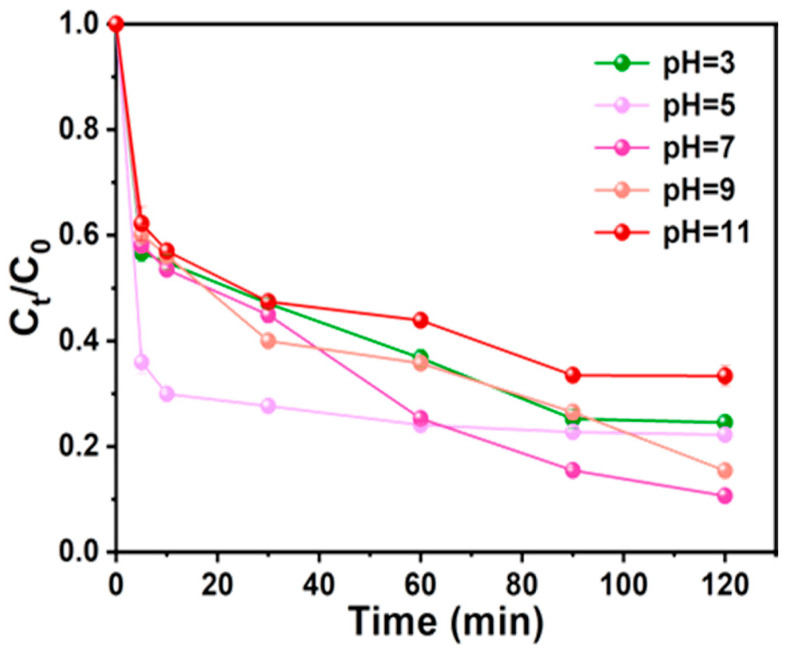
Effect of pH on Ph removal (Ph = 50 mg/L, MAC = 0.6 g/L, PS = 0.72 g/L, T = 25 °C).

**Figure 14 nanomaterials-15-00744-f014:**
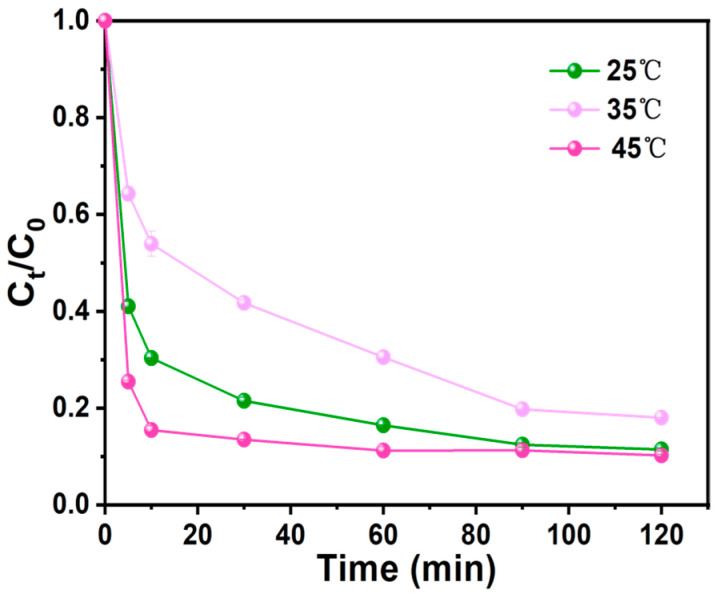
Effect of temperature on Ph removal (Ph = 50 mg/L, MAC = 0.6 g/L, PS = 0.72 g/L, pH = 7).

**Figure 15 nanomaterials-15-00744-f015:**
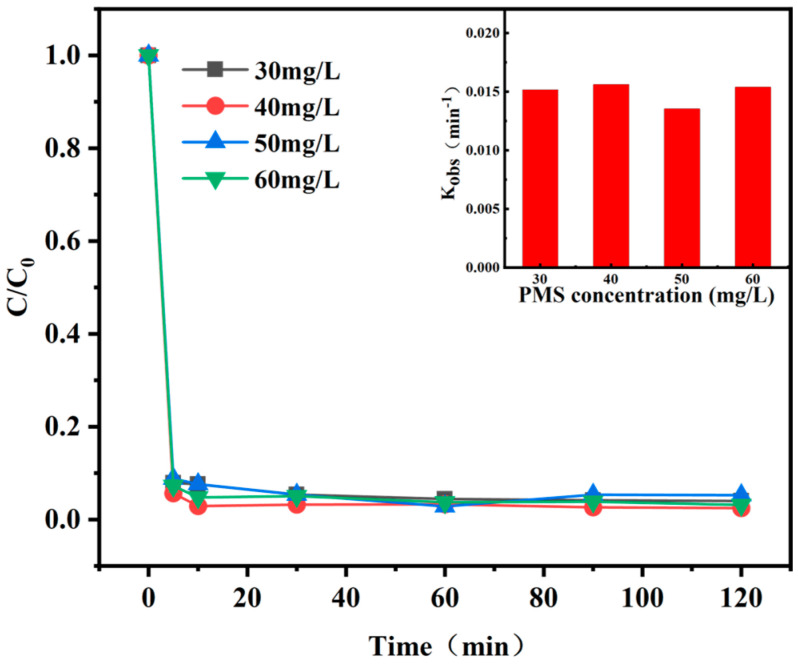
The effect of the initial concentration of Ph (MAC = 0.6 g/L, PS = 0.72 g/L, T = 25 °C, pH = 7).

**Figure 16 nanomaterials-15-00744-f016:**
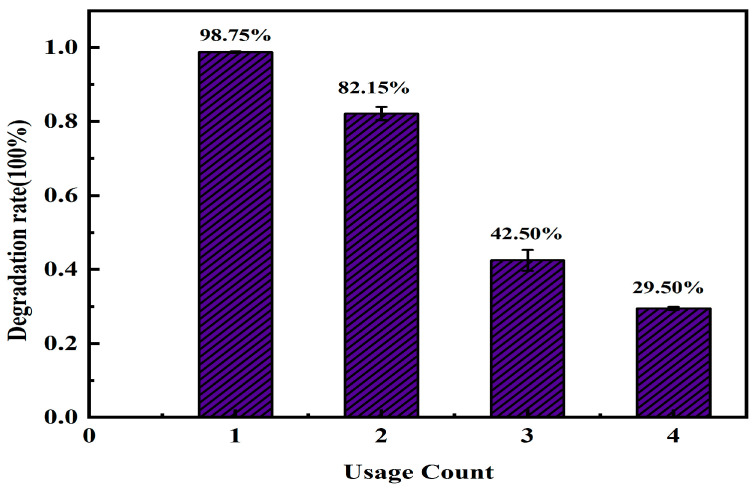
Ph removal tests for AC-K recycled over four cycles.

**Figure 17 nanomaterials-15-00744-f017:**
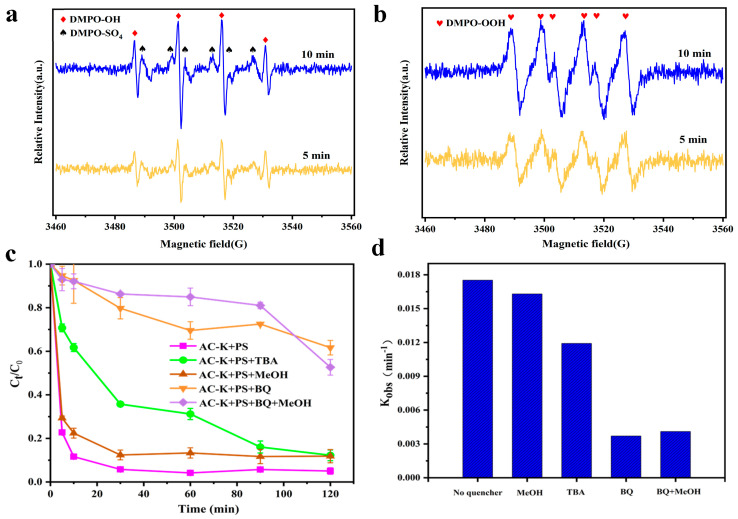
(**a**,**b**) EPR spectra of PS activation by AC-K; (**c**) Ph removal by the AC-K/PS system; and (**d**) reaction rate constants with and without radical scavengers.

**Figure 18 nanomaterials-15-00744-f018:**
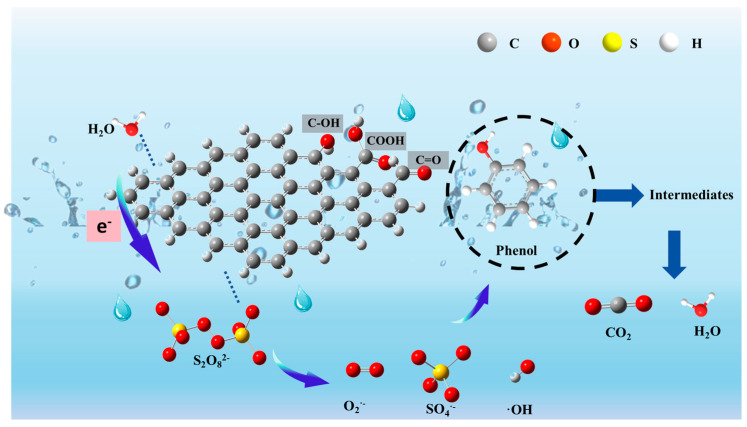
Free radical activation mechanism.

**Table 1 nanomaterials-15-00744-t001:** Pore structure parameters of AC, AC-Na, AC-K, and AC-K-De.

Samples	S_BET_ (m^2^/g)	V_t_ (cm^3^/g)	V_mic_ (cm^3^/g)	D_av_ (nm)
AC	462.8	0.456	0.085	3.9
AC-Na	426.3	0.383	0.098	3.6
AC-K	1468.8	0.861	0.527	2.3
AC-K-De	840.7	0.538	0.205	2.6

S_BET_: specific surface area; V_t_: total pore volume; V_mic_: micropore area; D_av_: average pore size.

**Table 2 nanomaterials-15-00744-t002:** Carbon crystalline structural parameters of AC, AC-Na, and AC-K.

Sample	*d*_002_ (Å)	*L_α_* (Å)	*L_c_* (Å)	*N*
AC	3.350	18.767	8.895	2.656
AC-Na	3.355	15.815	4.725	1.408
AC-K	3.400	10.362	2.586	0.761

**Table 3 nanomaterials-15-00744-t003:** O1S surface elements and functional group composition ratio in AC, AC-K, AC-Na, and AC-K-De.

Sample	Relative Content (%)				Relative Content (%)			
	C	O	K	Na	C=O	C=O	C-O-C	O-C=O
AC	91.91	8.09	-	-	29.3	41.0	21.0	8.7
AC-Na	91.98	7.8	-	0.22	3.0	15.8	57.1	24.1
AC-K	51.54	47.49	0.97	-	8.3	14.7	61.5	15.5
AC-K-De	72.38	21.98	5.16	-	23.6	30.4	25.3	20.7

**Table 4 nanomaterials-15-00744-t004:** Position and area of each O1S functional group in AC, AC-K, AC-Na, and AC-K-De.

Sample	Functional Groups	FWHM (eV)	Position (eV)	Area (%)
AC	C=O (carbonyl group)	2.00	531.03	3949
	C=O (ester amides)	1.89	532.21	5528
	C–O–C	2.16	533.30	2840
	COH, COOH	2.00	534.10	1160
AC-K	C=O (carbonyl group)	1.95	532.06	15,080
	C=O (ester amides)	1.52	533.67	26,710
	C–O–C	2.05	533.91	112,100
	COH, COOH	2.47	534.95	28,220
AC-Na	C=O (carbonyl group)	1.9	531.33	699
	C=O (ester amides)	2.8	532.19	3758
	C–O–C	2.0	533.40	13,570
	COH, COOH	2.29	534.99	5726
AC-K-De	C=O (carbonyl group)	2.41	529.24	64,360
	C=O (ester amides)	1.66	529.90	83,750
	C–O–C	1.33	530.75	69,080
	COH, COOH	1.66	531.56	56,540

**Table 5 nanomaterials-15-00744-t005:** C1S surface elements and functional group composition ratio in AC, AC-K, AC-Na, and AC-K-De.

Sample	Relative Content (%)				Relative Content (%)		
	C	O	K	Na	*sp*^2^-C/C=C	C–OH	C=O
AC	91.91	8.09	-	-	78.5	8.4	13.1
AC-Na	91.98	7.8	-	0.22	55.7	25.2	19.1
AC-K	51.54	47.49	0.97	-	65.0	15.8	19.2
AC-K-De	72.38	21.98	5.16	-	46.6	25.3	28.1

**Table 6 nanomaterials-15-00744-t006:** Position and area of each C1S functional group in AC, AC-K, AC-Na, and AC-K-De.

Sample	Types of Functional Groups	FWHM (eV)	Position (eV)	Area (%)
AC	*sp*^2^-C/C=C	1.12	284.81	38,550
	C–OH	1.34	286.10	4137
	C=O	2.28	287.73	6448
AC-K	*sp*^2^-C/C=C	1.15	284.78	57,300
	C–OH	1.50	286.23	13,890
	C=O	2.3	288.25	16,950
AC-Na	*sp*^2^-C/C=C	1.03	284.77	66,780
	C–OH	2.59	285.77	30,080
	C=O	2.40	288.10	22,930
AC-K-De	*sp*^2^-C/C=C	0.93	281.33	160,500
	C–OH	1.01	281.97	86,900
	C=O	1.83	282.89	96,890

## Data Availability

Data are contained within the article.

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
