# Peer review of "Activation of Persulfates Using Alkali-Modified Activated Coke to Promote Phenol Removal"

_nanomaterials, 2025, doi:10.3390/nano15100744_

Round 1

Reviewer 1 Report

Comments and Suggestions for Authors

Why was the manuscript sent to nanomaterials when there is no content related to nanomaterials at all?

Table 1.
Giving the specific surface area with an accuracy of cm2/g is rather incorrect. I think that the method used allows to determine it at +/-10m2/g. Similarly with the pore width.

In columns 3 and 4 it should be cm3/g and not m3/g.

How were the FTIR measurements performed?

How was the phenol concentration measured?

Pseudo-to-first-order, should be pseudo first-order.

There is no evidence of Ph degradation in the manuscript.

According to the caption of Fig. 10, in the work we are dealing with removal and not degradation.

Why did the authors not show the curve corresponding to equation 2 in Figs. 10 – 14 and 17c?

Why did the authors not include the values ​​of the rate constants (Kobs)?

(Fig. 14.) Why is the process at 25oC faster than at 35oC?

Fig. 14. does not correlate with the description in lines 381–385.

The mechanism of the process is unfounded because there is no evidence of phenol degradation.

Author Response

Dear Reviewers,

Thank you very much for the comments to our manuscript “Activation of persulfates using alkali-modified activated coke to promote phenol degradation”( nanomaterials-3567480). We thank you reviewers for your professional, insightful, and valuable comments. Each comment or remark has been studied carefully and correction or modification has been correspondingly made. For your convenience, the comments/remarks and the itemized responses with manuscript changes (highlights in yellow) are appended below. Thank you very much for your arduous work.

Best Regards.

Yours sincerely,

Dr. Yan Zhang

Reviewer 2 Report

Comments and Suggestions for Authors

Reviewer 1

Manuscript: nanomaterials-3567480

Title: Activation of persulfates using alkali-modified activated coke to promote phenol degradation

This paper examines carbon cokes (AC), as activated carbons, using different treatments (NaOH and KOH) and further activate them using persulfate (PS) for application in phenol degradation. It claimed that these AC can be important because they are industrial and agricultural waste materials. The paper is interesting and has results that may be published. The authors used a variety of characterization techniques (e.g., Adsorption of N2 at low temperature, XRD, FT-IR, XPS, ). However, it needs to be more organized on some points to make the key enhanced points clearer. The structural discussion is good but requires attention to always follow a specific order (e.g., start with AC and then AC-Na or AC-K) not crossing this order to facilitate the reading. Please, check and follow a consistent order. In addition, more details in the Materials and Methods section should be provided to anyone who can reproduce the experiments. Please, try to upload again Figures 1 and 4 to improve your resolution. It looks a little blurred. In general, check the formatting on the text. After revising the text, please also revise the Conclusions section. Below are the main points that require detailed revision.

Abstract

It should be rewritten to be clearer. There is an apparent contradiction here, for example: “AC-K proved to be a more effective activator than AC-Na, and was used for all subsequent experiments.” An above paragraph says the opposite!

  1. Introduction

In general, it should be checked for some language mistakes and to improve clearness.

  1. Materials and Methods

Line 81: Please, provide the purity of potassium persulfate (K2S2O8).

Line 90: “The sieved samples were then dried and stored in a dark …”. What was the temperature or other conditions for dry these samples?

Line 95: “The obtained dried potassium-modified AC sample (AC-K) was used…” Again, what was the further conditions for drying this sample?

Line 117: “For MAC recycling tests, the used MAC was washed alternatingly with deionized water and EtOH three times …” What was the mass used in this recycling? It was used only with spent material, or some fresh MAC was added.

Line 124: “water bath thermo-124 stat shaker (model and company details) set at 150 rpm.” The model and company details are missing.

Line 127: The same for suction filtration device.

Line 133: “and tested for data collection in the machine (EPR Bruker A300-10/12)”. Please, explain better this procedure (2.3.2.) and provide details of the equipment and measurements.

Line 140: Please, provide a consistent description of the equipment, model, manufacturer etc. for the characterizations of the materials (e.g., USA Micromeritics ASAP 2460 instrument; Bruker AXS, Germany). Also, describe the fundamental parameters used in each measurement for each technique.

  1. Result and Discussions

Line 164: Revise the connection between this line with the text that follows.

Table 1 and text (lines 166-178): correct the number of decimal places (figures) for SBET (maximum 1 decimal); The unit of pore volume does not seem correct. Is it not cm3/g? DAV (nm) should also be one decimal place. There is no such precision on those measurements.

Line 182 to 191: Do you have any literature to support such decomposition? There are some assumptions that may not be inferred from your experiments (AC treated under N2 atmosphere at a rate of 5°C/min up to 750°C) based only on the textural data. Looking quickly at the literature, I saw this paper (https://doi.org/10.1016/S0040-6031(98)00289-5) that may help to explain the thermal stability of K2CO3.

Line 192 to 195: I would recommend the discussion about the degradation process to be separated into a singular topic, to be clearer for the readers.

Line 196: If possible, the SEM image of AC should be at the same magnification (200 nm) of AC-Na and AC-K.

Line 213: “The XRD patterns revealed characteristic peaks of AC at 2θ = 22° and 43° corresponding to the (002) and (100) crystal planes of C, respectively (Fig. 3).” Does it correspond to some pdf file (database) or do you have a reference to cite?

Line 335: Could you justify the chosen initial conditions that you tested the degradation of Phenol?

Line 348: I believe that the written efficiency of phenol degradation could be reported with only one decimal place.

Line 398: “Although a small amount of AC-K was lost during the cleaning process,…”

Do you have any quantitative number for this loose of AC-K?

Line 432: “Ahmad et al.[42] demonstrated PS can be simultaneously activated by alkaline and phenolic oxides can under highly alkaline conditions (pH > 10.5).” Please, check it for a clearer statement. Also, check the discussion of the mechanism.

  1. Conclusion

Line 485: “AC-K exhibits significant morphological modifications, including a higher SSA and abundant microporous structure compared to plain AC, …” This statement cannot be inferred, since the microporous area of each AC was not determined (or reported).

Line 488: “Simultaneously, KOH modification dismantled the graphite crystal structure of AC…”. This term “dismantled” has been used in the whole text, but maybe there is a better word (scientifically speaking) to describe the change in the graphite structure.

Line 493: “AC-K exhibits high stability and common reusability.” This statement should be rewritten based on your results, saying what are the strong and weak points of this reutilization.

Comments on the Quality of English Language

The English could be improved to more clearly express the research.

Author Response

(The authors gave the same response as above.)

Reviewer 3 Report

Comments and Suggestions for Authors

This manuscript studied is Activation of persulfates using alkali-modified activated coke to promote phenol degradation. This work is interesting, which is a significant advancement over existing knowledge, but it needs improvements before considering for publication. The publication is recommended, subjected to revision as mentioned below in comments to the authors:

  1. Too many abbreviations are used. I recommend a nomenclature section for the abbreviations and variables used through the passage
  2. A graphical abstract are recommended in this paper.
  3. Please replace ‘’ mg/ml ‘’ by ‘’mg/mL’’, please check whole manuscript and revise it.
  4. Please replace ‘’ ‘’ by ‘’Figure.’’, please check whole manuscript and revise it.
  5. Please increase the size of Figure 2
  6. What are the innovative aspects of this study compared to other works?
  7. In the section 2. Alkali-modified AC preparation, please add the detail information such as sieve size, drying temperature and concentration of HCl, NaOH and KOH.
  8. In the section 3.2. Radical quenching experiments:
  • please add the detail information about the filtration such as the filter diameter;
  • Why did you adjust the pH solution to 6 before placing it in a dual-function water bath thermostat shaker?
  • Please add ultrasonic, machine (EPR Bruker A300-10/12), pH meter information such as model and city.
  1. The Figure 13 need more discussions.

Author Response

(The authors gave the same response as above.)

Round 2

Reviewer 1 Report

Comments and Suggestions for Authors

The comments and inaccuracies that influenced the decision are included in the attached pdf file.

Author Response

Dear Reviewers,

Thank you very much for the comments to our manuscript “Activation of persulfates using alkali-modified activated coke to promote phenol degradation”( nanomaterials-3567480). We thank you reviewers for your professional, insightful, and valuable comments. Each comment or remark has been studied carefully and correction or modification has been correspondingly made. For your convenience, the comments/remarks and the itemized responses with manuscript changes (highlights in yellow) are appended below. Thank you very much for your arduous work.

Best Regards.

Yours sincerely,

Dr. Yan Zhang

Issue 1:

Phenol standard curve  - incorrectly labeled axes.

Response:

Thank you for your valuable comments, we have made changes to the curve coordinates.

Details:

Issue 2:

The explanation for the relationship in Fig. 14 is confusing and not supported by any evidence.

Response:

Thank you for your suggestion. The mesoporous structure and surface functional groups (C-OH, defect sites) of the activated coke were not destroyed. Peroxynitrite (PS) interacts with the activated coke to form a surface-active complex, which oxidises the pollutant through a direct electron transfer process. This non-radical pathway dominates at low temperatures[35]. At 35°C, the rate of autolytic decomposition of PS is significantly lower than at 45°C, resulting in limited production of SO₄・- and ・OH, and the free radical pathway is unable to compensate for the loss of efficiency of the non-radical pathway due to the lower temperature.

Details:

  1. Tang, L.; Liu, Y.; Wang, J.; Zeng, G.; Deng, Y.; Dong, H.; Feng, H.; Wang, J.; Peng, B.; Enhanced activation process of persulfate by mesoporous carbon for degradation of aqueous organic pollutants: Electron transfer mechanism. Appl. Catal. B Environ. 2018,231, 1-10.

Issue 3:

The photographs in Fig. 2 show a size other than 2.9 – 4 nm.

Response:

Thank you for your comment, it was an oversight on our part and we have corrected it in the text!

Details:

In pages 7 of the revised manuscript:

Issue 4:

This is what the fit (lack of fit) of the kinetic curve according to the rate constant shown in Fig. 15 looks like. Red curve calculated for Kobs 0.015 min-1. Blue points copied from Fig. 15.

Response:

Thank you for your comments. We have derived our results based on Eq. (1).

                     (1)

k is the apparent primary rate constant for phenol removal;

C is the phenol concentration at different times (t);

C0 is the initial phenol concentration.

Issue 5:

Figs. 11 and 12 and 15. The curve (Ph=50mg/L, MAC=0.6 g/L, T=25 , pH=7) has a different course.

Response:

Thank you for your suggestion. Different batches of persulfate (PS) may have different activities due to small differences in synthesis conditions, and changes in laboratory temperature and humidity may affect reaction kinetics or catalyst surface adsorption behaviour. Defective sites or functional groups of the activated coke may oxidise during storage, leading to differences in activity between batches.

Reviewer 2 Report

Comments and Suggestions for Authors

Thank you for revising the manuscript. It is now acceptable for publication.

Author Response

(The authors gave the same response as above.)

Round 3

Reviewer 1 Report

Comments and Suggestions for Authors

Have a nice day and I wish you many scientific successes.